# The Diagnostic Landscape of Adult Neurogenetic Disorders

**DOI:** 10.3390/biology12121459

**Published:** 2023-11-22

**Authors:** Maggie W. Waung, Fion Ma, Allison G. Wheeler, Clement C. Zai, Joyce So

**Affiliations:** 1Division of General Neurology, Department of Neurology, Weill Institute for Neurosciences, University of California, San Francisco, CA 94158, USA; 2Institute for Human Genetics, University of California San Francisco School of Medicine, San Francisco, CA 94143, USA; 3Department of Neurology, University of Colorado Anschutz Medical Campus, Aurora, CO 80045, USA; 4Tanenbaum Centre for Pharmacogenetics, Molecular Brain Science, Campbell Family Mental Health Research Institute, Centre for Addiction and Mental Health, Toronto, ON M5T 1R8, Canada; 5Department of Psychiatry, Institute of Medical Science, Department of Laboratory Medicine and Pathobiology, University of Toronto, Toronto, ON M5S 1A8, Canada; 6Division of Medical Genetics, Department of Pediatrics, University of California, San Francisco, CA 94158, USA

**Keywords:** neurogenetic disease, whole genome sequencing, whole exome sequencing, complex neurological disease, adult-onset, genetic analysis, next generation sequencing, personalized medicine

## Abstract

**Simple Summary:**

This review provides a general overview of neurological genetic disorders that can emerge in adulthood. The goal is not to present an exhaustive list of adult-onset neurogenetic disorders, but instead to present a framework to help physicians recognize patterns of neurological disease that suggest a genetic cause. We discuss broad categories of neurological disease and the most common genetic etiologies in each category. We review common diagnostic approaches and pitfalls of current strategies. Whole-exome and whole-genome sequencing are emerging as more comprehensive tests for genetic disease, but it is still not clear how they should be applied to adult patients with complex neurological disease. This review highlights the need for more research to understand the diagnostic utility of genetic testing strategies and for increased collaboration between neurologists and geneticists.

**Abstract:**

Neurogenetic diseases affect individuals across the lifespan, but accurate diagnosis remains elusive for many patients. Adults with neurogenetic disorders often undergo a long diagnostic odyssey, with multiple specialist evaluations and countless investigations without a satisfactory diagnostic outcome. Reasons for these diagnostic challenges include: (1) clinical features of neurogenetic syndromes are diverse and under-recognized, particularly those of adult-onset, (2) neurogenetic syndromes may manifest with symptoms that span multiple neurological and medical subspecialties, and (3) a positive family history may not be present or readily apparent. Furthermore, there is a large gap in the understanding of how to apply genetic diagnostic tools in adult patients, as most of the published literature focuses on the pediatric population. Despite these challenges, accurate genetic diagnosis is imperative to provide affected individuals and their families guidance on prognosis, recurrence risk, and, for an increasing number of disorders, offer targeted treatment. Here, we provide a framework for recognizing adult neurogenetic syndromes, describe the current diagnostic approach, and highlight studies using next-generation sequencing in different neurological disease cohorts. We also discuss diagnostic pitfalls, barriers to achieving a definitive diagnosis, and emerging technology that may increase the diagnostic yield of testing.

## 1. Introduction

Patients with neurogenetic diseases can be first seen by the neurology generalist, or in any neurology subspecialty clinic, as they may embody atypical presentations of acquired neurological disease that involve multiple neurological domains. A major challenge in the diagnosis of neurogenetic disease is a lack of access to neurogenetic clinical expertise for many patients [1,2]. In addition to clinical barriers to diagnosis of neurogenetic disease, the costs of genetic testing, particularly whole-exome sequencing (WES) and whole-genome sequencing (WGS) are often not covered by insurance for adult patients [3,4,5,6]. Moreover, there is phenotypic heterogeneity in these diseases, as mutations in the same gene can cause distinct clinical syndromes in different individuals. For example, *SCN2A* mutations that lead to gain of function or loss of function of the encoded sodium voltage-gated channel Nav1.2 can manifest as a spectrum of phenotypes, such as seizure disorders [7], intellectual disability [8], autism spectrum disorder [9], episodic ataxia [10], and schizophrenia [11], with different phenotypic combinations, including one report of overlap of all phenotypes in the same individual [12]. As another example, the hexanucleotide (GGGGCC) repeat expansion in the non-coding region *C9orf72* can cause both frontotemporal dementia (FTD) and/or amyotrophic lateral sclerosis (ALS) [13,14]. Likewise, genetic allelic or locus heterogeneity can be appreciated when the same clinical syndrome can result from different pathological variants in the same gene or different genes. For example, mutations in *TBP* (spinocerebellar ataxia type 17; SCA17), *ATN1* (dentatorubral-pallidoluysian atrophy; DRPLA), *JPH3* (Huntington disease-like 2; HDL2), *C9orf72* (Frontotemporal dementia and/or Amyotrophic lateral sclerosis; FTD/ALS1), and *FXN* (Friedreich ataxia) can all cause Huntington’s disease-like syndromes [15].

Features that should alert the clinician to consider a neurogenetic evaluation include (1) family history of similarly afflicted individuals, (2) atypical onset of disease, (3) involvement of multiple neurological domains or systemic features, and (4) subacute or chronic, progressive, and unrelenting course. The presence of consanguinity may also be helpful in assessing the utility of genetic testing for autosomal recessive (AR) disorders, and some neurogenetic disorders are found more frequently in certain ethnic groups. However, a lack of family history does not preclude a neurogenetic diagnosis due to several factors, including de novo mutations, repeat expansion disorders or incomplete penetrance of disorders. Furthermore, multiple family members affected with distinct neurological disorders may also represent a red flag given that many genes associated with neurological disorders can present with multiple phenotypes. For example, mutations in *VCP* can be variably associated with Charcot-Marie-Tooth disease (CMT), FTD, ALS, and/or inclusion body myopathy with early-onset Paget disease in different members of the same family [16,17,18]. 

In a study of 1411 patients with unexplained adult-onset neurological disorders, multi-gene panel testing for 7 different categories of neurological disease (ataxia and spasticity, leukoencephalopathy, movement disorders, neurodegeneration with brain iron accumulation, paroxysmal episodic disorders, progressive myoclonic epilepsy, and ALS) revealed a diagnostic yield of 10% [19]. The category with the highest diagnostic yield was the ataxia and spasticity group, followed by the paroxysmal episodic disorders and leukoencephalopathy panels [19]. WES, which can capture all exons of all known genes, and WGS, which analyzes the entire human genome with the potential to detect copy number variants, non-coding variants, and repeat expansions, are predicted to further increase the diagnostic yield of adult-onset neurogenetic disorders. Periodic re-analysis of WES and WGS can further increase the diagnostic yield as new disease-causing genes and variants are identified [20,21,22]. Already, WES and WGS have demonstrated increased diagnostic yield in pediatric neurogenetic conditions [23,24,25] and likely will demonstrate a higher yield in adult neurogenetic disorders, though this is yet to be definitively determined.

## 2. Neurogenetic Syndromes

### 2.1. Movement Disorders: Parkinson’s Disease, Dystonia, Ataxia, Spastic Paraparesis

Many neurogenetic disorders affect movement beginning in adulthood. The literature suggests that WES/WGS has a diagnostic yield that is comparable to or higher than targeted sequencing of relevant genes using gene panels in hereditary movement disorders such as Parkinson’s disease (PD), dystonia (DYT), and ataxia/spastic paraplegia [26].

Monogenic PD represents a small fraction of PD cases, but most pathogenic variants are associated with early-onset PD before 40 years of age. With WES, 11.3% [27] to 14% [28,29] of patients with early-onset PD can be found to have a pathogenic variant compared with targeted panels with a yield ranging from 0% [30] to 4.3% [31], with targeted panels yielding a much lower rate due to the use of gene panels lacking the most updated complement of PD-associated genes.

Hereditary DYT most often begins in childhood but can emerge in adolescence and adulthood. In a study examining genetic testing of both pediatric and adult patients with DYT, 12% of individuals were found to have pathogenic or likely pathogenic variants identified with WGS, with earlier age of onset, younger age at testing, and a combined DYT phenotype with other movement disorders more likely to yield a genetic diagnosis [32].

Hereditary ataxias are a heterogeneous and complex group of disorders characterized by cerebellar ataxia and oculomotor abnormalities, often leading to gait impairment, and speech and swallowing difficulties. Additional features of spinocerebellar ataxias (SCAs) include retinopathy, optic atrophy, peripheral neuropathy, extrapyramidal symptoms, epilepsy, and cognitive dysfunction. Ataxic syndromes are a major reservoir for neurogenetic disease. The prevalence of autosomal dominant (AD) cerebellar ataxias is estimated to be about 2.7 per 100,000 and 3.3 per 100,000 for AR cerebellar ataxias [33]. Most SCAs have a typical age of onset in adulthood, usually in the third or fourth decade. The more common SCAs (SCA1, SCA2, SCA3, SCA6, SCA7, SCA17, DRPLA) are classically caused by CAG trinucleotide repeat expansions. SCA3 (Machado–Joseph disease) is the most common AD ataxia and Friedreich ataxia is the most frequent AR ataxia [33]. More recently, intronic repeat expansions in *RFC1* (cerebellar ataxia, neuropathy, and vestibular areflexia syndrome, CANVAS) [34] and *FGF14* (SCA27B) [35] have been identified as more significant causes of adult-onset ataxia than previously appreciated. Fragile X-associated tremor/ataxia syndrome (FXTAS) is an X-linked late-onset degenerative disorder in Fragile X disease premutation carriers of a CGG repeat expansion in *FMR1*.

Hereditary spastic paraplegias (HSP) are a group of disorders characterized by a gradual worsening of stiffness and spasticity beginning in the legs due to corticospinal tract degeneration. The prevalence of AD and AR HSP are both about 1.8 per 100,000, with spastic paraplegia, type 4 (SPG4), and SPG11 being the most common types of AD and AR HSP, respectively [33]. Estimates of prevalence rates do not include X-linked or other inherited forms of these conditions and, although rare, potentially underestimate the prevalence of genetic hereditary ataxias and HSP. Classically, HSP has been divided into two major categories, the pure (uncomplicated) phenotype of spastic paraplegia alone or the complex (complicated) phenotype associated with additional neurological and extra-neurological manifestations such as cognitive impairment, peripheral neuropathy, cerebellar signs, and eye signs. While clinically useful, this classification has limited value in guiding the molecular screening of HSP as the same mutation can lead to both pure and complex forms of HSP. Moreover, WES and WGS diagnostic approaches have revealed genetic overlap between HSP and other neurogenetic conditions, more commonly inherited ataxias, neuropathies, and motor neuron diseases, found to be associated with genes originally described in HSP patients [36]. In this case, both the site and type of the mutation might have distinct pathogenic effects on the gene product, which might explain some of the underlying differences in their clinical presentation.

### 2.2. Epilepsy and Intellectual Disability

Epilepsy is one of the most common neurological diseases globally, with genetic factors thought to play a role in a significant number of these patients. In a cross-sectional study of 2008 adult patients who underwent a multigene epilepsy test, 10.9% received a diagnostic finding [37]. Genetic epilepsies were identified in adults with child-onset epilepsy, with the most common variants in *SCN1A*, *KCNT1,* and *STXBP1*, as well as adults with adult-onset seizures with gene variants in *FLNA* and *LGI1*, which are associated with reduced penetrance and variable expressivity [37].

For adults presenting with epilepsy of unknown etiology, features that make monogenic epilepsy more likely include seizure onset in infancy [37], comorbid intellectual disability (ID) or developmental delay (DD) [37,38,39], and pharmacoresistant epilepsy [37]. Genetic testing in an adult cohort of patients with epilepsy revealed pathogenic variants could be missed in smaller gene panels, particularly in patients with early DD and late seizure onset [40]. In a large systematic review examining the diagnostic yield of different genetic testing modalities in all patients with epilepsy, WGS returned the highest yield (48%) compared to WES (24%), multigene panels (19%), and chromosomal microarray (9%) [41]. While most genetic epilepsies are associated with generalized seizures, several genes have been identified as causative of focal epilepsies, such as *NPRL2*, *NPRL3* [42], and *DEPDC5* [43], genes involved in regulating the mammalian target of rapamycin (mTOR) pathway, as well as *FLNA* (focal seizures due to unilateral or bilateral periventricular nodular heterotopia) and *LGI1* (focal seizures with auditory features). Epilepsy can also be a common manifestation of mitochondrial disease due to mutations in mitochondrial DNA and nuclear-encoded genes that affect mitochondrial function and maintenance, such as *POLG* [44].

### 2.3. Neuromuscular Disorders

Hereditary neuropathies, comprising CMT, also known as hereditary motor/sensory neuropathy (HMSN), distal hereditary motor neuropathy (HMN), hereditary sensory and autonomic neuropathy (HSAN), hereditary neuropathy with liability to pressure palsies (HNPP), and hereditary brachial plexus neuropathy (HBPN), often become clinically apparent or begin in adulthood. Neuropathies can also present as the initial or prominent feature in complex neurogenetic diseases such as familial amyloid polyneuropathies, hereditary ataxias with neuropathy, and complicated HSP. Acquired neuropathy may be superimposed on an inherited neuropathy since those with underlying inherited neuropathy are more susceptible to injury from other causes of nerve injury [45]. Genetic overlap and phenotypic diversity of hereditary neuropathies are complex, including phenotypic overlap with axonal CMT, ataxias, distal myopathies, and HSP, making the selection of genes for testing difficult, but genome-wide sequencing is poised to change the diagnostic approach for these disorders.

The group of inherited neuropathies collectively known as CMT is the most common inherited neuromuscular disease, with a prevalence in the general population varying widely by region and race/ethnicity from 9.4–82 per 100,000 [46]. The classic form of CMT is characterized by slow and progressive distal weakness, sensory loss, and foot deformities, particularly pes cavus and hammertoes. Patients with CMT often do not have sensory or pain symptoms, so clinical presentation, and thus diagnosis, can be delayed until adulthood.

The classification of CMT, which previously prioritized electrophysiologic categories of motor nerve conduction velocities (demyelinating = CMT1, axonal = CMT2, dominant intermediate = DI-CMT), is now evolving to a multi-tiered classification including inheritance pattern, phenotype, and genetic designation [47]. Nerve conduction studies remain helpful in distinguishing between subtypes of different genetic causes of CMT, non-CMT inherited neuropathies and acquired chronic neuropathies.

CMT has heterogeneous modes of inheritance and genetic causes, with mutations in *PMP22*, *GJB1*, *MPZ*, and *MFN2* representing over 90% of genetically defined CMT1 [48]. Peripheral myelin protein 22 is a critical membrane glycoprotein component in compact myelin, and *PMP22* duplication and deletion copy number variants (CNVs) are the most common cause of inherited demyelinating neuropathy, accounting for approximately 78% of cases [48]. Furthermore, a CNV on chromosome 17p12 in the intronic region upstream of *PMP22* has also been described as causative for CMT1A [49], likely through regulation of *PMP22* transcript expression [50]. Testing for this region is not included in typical neuropathy gene panels, and identification of this variant would be missed on WES.

While *PMP22* duplication variants are typically associated with CMT1A, *PMP22* deletions, leading to *PMP22* haploinsufficiency, are associated with HNPP, characterized by recurrent, painless, focal sensory motor neuropathies triggered by mechanical stress on the peripheral nerves. HNPP usually begins in the second or third decade but can present much later. It is less common than CMT1A with a prevalence estimated at 1–7 per 100,000 [51,52], but prevalence rates could be much higher given milder presentations and likely under-diagnosis [53].

CMT2, characterized by axonal neuropathy, typically begins later, usually in the second or third decade of life. X-linked CMT, typically due to *GJB1* mutations, can present with frequent falls in early adulthood with weakness, muscle atrophy, and areflexia, with females exhibiting later onset. Even in a cohort of unexplained axonal neuropathies in the middle-aged and elderly, 18.3% of patients had disease-causing variants identified by WES [54].

As the number of CMT genes corresponding to the different subtypes continues to expand, the likelihood of identifying a causative genetic variant is increasing. In a retrospective, single-center study, gene panel testing of 108 patients with CMT identified 17 patients (15.7%) with pathogenic or likely pathogenic variants [55]. In a nationwide laboratory study in Japan, the combination of microarray, gene panel sequencing, and WES of 2598 cases identified pathogenic or likely pathogenic variants in 798 patients (30.7%) [56].

Regarding muscle disorders, myotonic dystrophies (DM) are the most common muscular dystrophies in adults, characterized by progressive muscle weakness, myotonia, cardiac conduction abnormalities, and non-muscular features that can include cataracts, sleep, cognitive, behavioral, and gastrointestinal disease. The prevalence of DM is estimated to be 10 per 100,000, with DM1 estimated to be 4 times more prevalent than DM2 [57]. DM1 and DM2 are both repeat expansion disorders in non-coding regions; DM1 is caused by CTG repeat expansion in the 3′-untranslated region of *DMPK* and DM2 is caused by CCTG repeat expansion in intron 1 of *CNBP*. Both disorders are inherited as an autosomal dominant trait and show a wide phenotypic variability, mostly related to the meiotic and mitotic instability in tissues which characterizes related CTG and CCTG pathological expansions.

Other hereditary muscular dystrophies that have a wide age range of muscle weakness onset include dystrophinopathies (Duchenne and Becker muscular dystrophy, allelic variations due to mutations in *DMD*), facioscapulohumeral muscular dystrophy (due to hypomethylation of *DUX4* arising from either a combination of *D4Z4* contraction with a permissive haplotype or a mutation in *SMCHD1*), and limb-girdle and distal muscular dystrophies (associated with mutations in multiple genes). Oculopharyngeal muscular dystrophy (OPMD) is caused by GCN repeat expansion in *PABPN1* and presents with asymmetric ptosis and dysphagia that only manifests in adulthood, usually presenting in the fourth to sixth decade. Collagen VI-related dystrophies, like Bethlem muscular dystrophy, are often diagnosed in adulthood.

MNDs, exemplified by ALS, affect an estimated 5 per 100,000 individuals [58], with the most common genetic etiology, a hexanucleotide repeat expansion in *C9orf72*, accounting for 10% of patients with clinical ALS that can also be associated with FTD. Pathogenic variants in *SOD1*, *FUS,* and *TARDBP*, among many others, can also lead to ALS. Spinal and bulbar muscular atrophy (also known as Kennedy’s disease) is due to an X-linked recessive CAG repeat expansion in the *AR* gene.

Phenotypic classification is helpful but cannot fully predict the genetic basis of neuromuscular disorders, as defects in one gene can cause different phenotypes (phenotypic heterogeneity), and the same phenotype can be caused by pathogenic variants in different genes (locus/genetic heterogeneity). Inherited diseases of muscle are frequently observed in neurometabolic and mitochondrial disease, for which they most often present with exercise-induced weakness and/or muscle pain.

### 2.4. Cognitive Neurodegenerative Disease

Although Alzheimer’s disease (ALZ) has a strong genetic component, most patients with ALZ are thought to have a polygenic mechanism of disease. Approximately 95% of all ALZ is late-onset (>65 years of age), with about 5% representing early-onset ALZ (<65 years of age) [59]. While early-onset ALZ is more likely to have a monogenic cause, it is not synonymous with genetic ALZ. Mendelian AD ALZ (caused by mutations in *APP*, *PSEN1*, or *PSEN2*) is rare, explaining only 5–10% of early-onset ALZ cases [59].

FTD, characterized by frontotemporal lobar degeneration that affects personality, behavior, and language has an estimated prevalence of 15–22 per 100,000 [60]. FTD is highly heritable, with 42% of patients with a positive family history of dementia [61]. The most common genetic cause of FTD is the hexanucleotide (GGGGGCC) repeat expansion in the non-coding region of the *C9orf72* gene, which accounts for up to 12% of familial and 3% of sporadic FTD and can also be associated with ALS [62]. Mutations in *MAPT* and *GRN* are also commonly found in familial FTD and similarly account for ~3% of sporadic FTD.

The incidence of prion disease is estimated to be about 0.2 per 100,000 annually, with genetic prion disease related to *PRNP* mutations accounting for 10% of this number [63]. Genetic prion disease is characterized by 3 clinical syndromes: Genetic Creutzfeldt–Jakob disease with rapidly progressive dementia; Gerstmann–Straussler–Scheinker disease with predominantly ataxia, and fatal familial insomnia with dysautonomia and severe sleep disturbance.

### 2.5. Leukodystrophies and Other Diseases of White Matter

Inherited disorders affecting cerebral white matter can be hypomyelinating or demyelinating, with differences detectable on MR imaging. Symptoms of some forms of leukodystrophies have onset during infancy or childhood; however, attenuated phenotypes of such forms may present in adulthood (metachromatic leukodystrophy, Krabbe disease, Alexander disease), whereas, on the other hand, other forms of inherited leukodystrophies are typically characterized by an adult-onset (vanishing white matter disease, X-linked adrenoleukodystrophy) [64]. The most common symptoms include lower extremity weakness, cognitive dysfunction, mood, and behavior changes. Gait ataxia, sensory symptoms with autonomic dysfunction, as well as extrapyramidal movement disorders and seizures, can be accompanying features. Extra-neurological symptoms include cataracts, optic nerve atrophy, endocrine dysfunction, polyneuropathy, hypodontia, cutaneous signs, and gastrointestinal dysfunction.

The frequency of leukodystrophies is estimated to range from 2–13.3 per 100,000 [65,66]. A diagnostic algorithm proposed by Köhler et al. includes (1) identifying clinical syndromes or MRI findings suggestive of leukodystrophy, (2) excluding acquired vasculopathies, toxic, inflammatory, neoplastic, and degenerative causes of white matter disease, (3) acquiring additional information such as family history and using pattern recognition of clinical presentation, (4) performing biochemical testing, WES/WGS, and considering brain biopsy [67]. WES may increase the diagnostic yield of suspected adult-onset leukodystrophies from ~50% to 72% [68], arguing for the use of WES as a first-line diagnostic test.

The major differential diagnosis of leukoencephalopathies includes the inherited cerebral vasculopathies, such as cerebral autosomal dominant arteriopathy with subcortical infarcts and leukoencephalopathy (CADASIL), caused by *NOTCH3* mutations. A number of other genetic syndromes of cerebral small vessel disease with variable anatomic distributions of pathological cerebrovascular (ischemic and hemorrhagic) findings on imaging exist, including cerebral autosomal recessive arteriopathy with subcortical infarcts and leukoencephalopathy (CARASIL), cathepsin-A-related arteriopathy with strokes and leukoencephalopathy (CARASAL), *COL4A1*-related disease, pontine autosomal dominant microangiopathy and leukoencephalopathy (PADMAL), Fabry disease, *HTRA1* heterozygotes, pseudoxanthoma elasticum, hereditary cerebral hemorrhage with amyloidosis (HCHWA), and retinal vasculopathy with cerebral leukodystrophy and systemic manifestations (RVFL-S), among others. Clinical clues to monogenic stroke syndromes include recurrent stroke, age at onset younger than 50 years old, lack of stroke risk factors, positive family history, and characteristic neuroimaging findings, most commonly symmetrical and progressive periventricular and subcortical white matter hyperintensities [69]. Other neurological features such as cognitive impairment, migraines, and mood disturbances may accompany cerebrovascular disease in these disorders, and systemic findings such as skin abnormalities, cardiac disease, diabetes, and kidney disease may help indicate specific disease categories.

### 2.6. Neurometabolic Diseases

NMDs are a heterogeneous group of genetic disorders with alterations in cellular metabolism, often due to enzyme deficiency that causes substrate accumulation leading to toxic effects, or lack of downstream product synthesis, such as neurotransmitter synthesis disorders. Due to the mechanism of NMDs, there are often targeted therapies and strategies to reduce the deleterious effects of toxic accumulation and/or supplement deficient enzymes or substrates that can improve the clinical course of NMDs, making it critical to identify these diseases as early as possible.

More common NMDs will be mentioned here; for more comprehensive reviews of treatable adult-onset NMDs, see [70,71]. Broad categories of NMD that can present in adulthood include urea cycle disorders (such as ornithine transcarbamylase deficiency), late-onset forms of lysosomal storage disorders (such as Niemann–Pick disease type C, Fabry disease, and Pompe disease), peroxisomal disorders (such as X-linked adrenoleukodystrophy), mineral metabolism disorders (such as Wilson disease), remethylation disorders (such as disorders of intracellular cobalamin metabolism, the most common of which is cobalamin C deficiency) and heme synthesis disorders (such as the hepatic porphyrias). Clinical features of these disorders are variable and can manifest with overlapping phenotypes. The most common neurological signs in adult patients diagnosed with inherited metabolic disorders are extrapyramidal/cerebellar signs, cognitive dysfunction, and myelopathy [71]. Peripheral neuropathy, epilepsy, psychosis, myopathy, and optic neuropathy can also be isolated manifestations of NMD or develop in combination [70,71].

Inherited mitochondrial disorders are a subset of NMD with a prevalence rate in adults of 20 per 100,000 [72]. Most mitochondrial disorders are attributable to mitochondrial DNA mutations compared to about 15% of patients with mitochondrial disorders who have nuclear mitochondrial mutations [72]. Classical mitochondrial syndromes presenting in adulthood include subacute, painless, and progressive vision loss (as in Leber hereditary optic neuropathy, LHON), stroke-like episodes associated with headache (as in mitochondrial myopathy, encephalopathy, lactic acidosis, and stroke-like episodes, MELAS), myoclonic epilepsy (as in myoclonic epilepsy associated with ragged-red fibers MERRF), neuropathy and ataxia alongside retinal deterioration (as in neuropathy, ataxia, and retinitis pigmentosa, NARP), ophthalmoplegia with cardiomyopathy (as in Kearns–Sayre syndrome, KSS) and bilateral, symmetric ptosis and ophthalmoplegia (as in chronic progressive external ophthalmoplegia, CPEO). Mitochondrial myopathies in general manifest with fatiguability, exercise-induced myalgia, exercise intolerance, and lactic acidemia. Additional features such as young onset sensorineural hearing loss, optic neuropathy, cerebellar ataxia, sensory ataxia, peripheral neuropathy, migraine, cognitive impairment, spasticity, and extrapyramidal movement disorders can be features of mitochondrial disease, whose presence and predominance in the phenotype can depend on the underlying mutation.

The time course of NMD can be variable and be acute, episodic, or slowly progressive. Features include a positive family history, parental consanguinity, a very chronic course, or atypical features of a common etiology. Additionally, changes in basal metabolism like systemic illness, exercise, or fasting states can precipitate acute neurological symptoms. Oftentimes, there is accompanying multi-organ dysfunction associated with these cases, while additional audiological, ophthalmological, and imaging assessments can be helpful. Organomegaly (splenomegaly, hepatomegaly) can point to abnormal storage of substrates or metabolites.

A proposed diagnostic algorithm for identifying genetic NMD in adults incorporates clinical symptoms with recognition of specific MR imaging patterns, then expanded phenotypic testing with brain and abdominal imaging, EMG/NCS, hearing, and ophthalmological assessment, followed by targeted biochemical testing [70].

### 2.7. Episodic Neurologic Syndromes-Paroxysmal Movement Disorders, Episodic Ataxia, Hemiplegic Migraine

Episodic neurologic syndromes are relatively common and can be challenging to diagnose. While genetic episodic neurologic syndromes are rare, they may become more apparent as more genes and mutations are described. Symptoms may also not be recognized as significant until they have progressed in severity and resulted in chronic or permanent neurological damage.

Most paroxysmal movement disorders present in childhood, though paroxysmal eye movements, paroxysmal kinesigenic dyskinesia (PKD), and paroxysmal exercise-induced dyskinesia (PED) can present in adolescence or early adulthood. PKD is a rare condition with an estimated prevalence of 0.7 per 100,000 [73] that is characterized by brief (<1 min) attacks of chorea and/or dystonia triggered by sudden voluntary movements, while PED attacks are triggered by prolonged exercise and last for minutes to hours. Some genes that cause paroxysmal disorders are involved in ion channel function, however, not all paroxysmal disorder-associated genes are directly involved in synaptic transmission (e.g., *PRRT2*, *SLC2A1*).

Episodic ataxias (EAs) are also rare disorders with an estimated prevalence of less than 1 per 100,000 [74]. The onset of EAs is typically in childhood, though late onset even up to the fifth or sixth decade has been reported with EA type 2. Attacks of ataxia along with dysarthria, tremor, diplopia, nystagmus, dystonia, hemiplegia, headache, and tinnitus may last for a few seconds up to several days. Episodes can be triggered by alcohol, caffeine, systemic illness, stress, startle, or strong emotions. Over time, a mild progressive cerebellar ataxia may develop.

Hemiplegic migraine is characterized by attacks of headache with aura symptoms and unilateral motor weakness lasting up to 72 h. The prevalence of hemiplegic migraine is estimated to be 10 per 100,000 [75]. Familial hemiplegic migraine is autosomal dominantly inherited with mutations found in three ion transporter genes, *CACNA1A* (which has phenotypic overlap with EA type 2), *ATP1A2*, and *SCN1A*.

Rarely, inherited muscle channelopathies like hypokalemic periodic paralysis and hyperkalemic periodic paralysis can begin with attacks in early adulthood. This heterogeneous group of disorders is characterized by episodic attacks of muscle weakness occurring with variable frequency and lasting for minutes to hours, but sometimes up to days. Some patients eventually develop a chronic progressive myopathy. Common triggers include low temperature and emotional stress, with potassium-rich food, post-exercise rest triggering hyperkalemic periodic paralysis and excessive salt intake, a lack of exercise, and alcohol consumption precipitate hypokalemic periodic paralysis.

## 3. Current State and Challenges of Neurogenetic Testing

The most common neurogenetic testing options include chromosomal microarray (CMA), gene panels, repeat expansion testing, biochemical testing, and WES. An illustrative example of the current approach to neurogenetic testing is shown in Figure 1. The most common, widely available first-line genetic testing strategies for the diagnosis of neurogenetic disorders are shown in Table 1.

### 3.1. Chromosomal Microarray

CMA is a probe-based technology that detects CNVs (deletions and duplications) across the genome. The American College of Medical Genetics and Genomics (ACMG) recommends CMA as a first line of testing in determining a genetic cause for unexplained ID, DD, or autism spectrum disorder, as well as in patients with congenital anomalies [76]. While CMA can be a powerful diagnostic tool when used appropriately, the test is limited to the detection of large deletions or duplications, and CNVs smaller than 100–200 Kb are not well-detected. CMA also does not detect point mutations, structural chromosomal rearrangements, or other variant types.

### 3.2. Gene Panel Testing

Gene panel testing simultaneously sequences a curated collection of genes that are known to cause certain diseases or phenotypes. This type of testing is particularly helpful when a patient’s phenotype(s) or suspected condition is associated with variants in many genes. Gene panels can be updated as new conditions are described and new genes are linked to specific phenotypes, but this requires repeat testing over time. Moreover, the genes included in a panel are not standardized across clinical testing laboratories [77]. This lack of standardization may lead to missed testing of genes that would have provided a diagnosis, thus extending the patient’s diagnostic odyssey. The opposite may also happen where genes that are not yet known to cause the suspected condition are included in the panel and result in inconclusive or conflicting outcomes. Additionally, the differences in gene panels between laboratories may cause confusion and uncertainty when choosing a test, especially for non-expert clinicians, and can lead to differences in standards of patient care.

### 3.3. Repeat Expansion Testing

Repeat expansion disorders (REDs) have distinguishing clinical features, but a high index of suspicion is needed to direct testing for repeat expansions. The age of onset of clinical symptoms is often negatively correlated to the sizes of the repeat expansions (e.g., SCA17, DRPLA, SCA2, SCA3, OPMD), with lower repeat expansions associated with disease onset in the 60s and 70s. In addition, lower repeat expansions or intermediate alleles may give rise to reduced penetrance for a number of neurodegenerative diseases, including SCA17 [78], OPMD [79], FXTAS [80], ALS [81], SCA3 [82], and Huntington’s disease [83]. However, as a highly specialized test, there are limitations to repeat expansion testing. By targeting specific conditions, RED testing is often done as a single gene test or, in rarer cases, as a small panel test. For example, diagnosis of DM1 typically requires PCR, with an additional Southern blot analysis needed to detect larger expansions. Diagnosis of DM2 often requires PCR repeat-primed assay in addition to these. The detection of repeat expansions is limited in WES, but REDs could be screened for using bioinformatic methods for repeat expansions with WGS [84,85].

Testing for certain REDs is significantly limited by the availability of laboratories that can perform these tests, reducing the accessibility of repeat expansion testing. For example, Friedreich ataxia *(FXN),* CANVAS, and SCA27B present unique diagnostic challenges, as they are most often caused by intronic repeat expansions that require dedicated validation of assays. The biallelic AAGGG intronic repeat expansion in *RFC1* associated with CANVAS and the deep intronic GAA repeat expansion in *FGF14* associated with SCA27B are absent from most commercial repeat expansion panels for ataxia, despite emerging evidence demonstrating their prevalence as some of the more common causes of undiagnosed cerebellar ataxia [86,87] and late-onset cerebellar ataxia [35,88]. *RFC1*-related disease is also increasingly being recognized as a cause of idiopathic chronic sensory neuropathy [89,90] in adults.

### 3.4. Biochemical Testing

When there is a suspicion for an NMD, biochemical testing often includes serum, urine, or CSF testing to examine the accumulation or depletion of metabolites in enzymatic pathways affected by genetic mutations, or to directly measure levels or activity of specific enzymes. For example, Gaucher disease, an inherited lysosomal storage disorder, can be tested for by measuring levels of the beta-glucosidase enzyme in leukocytes. Specific tests can screen for various types of metabolic disorders, including lysosomal or glycogen storage disorders, mitochondrial diseases, peroxisomal disorders, and metal metabolism disorders, among others. Molecular genetic testing can also be used for diagnosing metabolic disorders and determining carrier status and is often done as single gene tests or panel tests. Molecular testing can offer an expedited route to the diagnosis of NMDs, but negative molecular testing does not exclude these diagnoses and may still necessitate specialized biochemical testing. Testing for mitochondrial disorders can be more complex, as mitochondrial mutations may be confined to certain tissues in the body (known as heteroplasmy) and may not be detected if an unaffected tissue is tested. For example, mitochondrial myopathies are often diagnosed by mitochondrial genome analysis from muscle biopsy of an affected muscle group but may not be diagnosed by testing on a blood sample or other tissue.

### 3.5. Whole-Exome Sequencing

WES is the largest, most comprehensive genetic test currently available through clinical testing. By sequencing all protein-coding regions of DNA, this phenotype-driven test analyzes thousands of genes in addition to those found on gene panels. This also provides optional reporting of secondary findings for medically actionable conditions as defined by ACMG guidelines [91]. The additional coverage can be particularly helpful in complex cases, as WES is commonly used when first-line genetic testing is inconclusive. However, as a large-scale test, it is not uncommon for a larger number of identified variants of uncertain significance that require further clarification of their clinical impacts. This problem is often remedied by pairing the test with parental or other family member testing for segregation of the identified variants but is less utilized for adult patients as parental samples are often not easily available.

## 4. The Future of Neurogenetic Testing

Accurate and efficient diagnosis of neurogenetic disorders is required for clinically relevant precision management of affected patients, including genetically targeted therapies for an increasing number of Mendelian disorders [92,93] and enabling patients to enroll in clinical trials relevant to their disease. Standard-of-care genetic testing in most individuals with complex neurological conditions comprises multiple steps in a lengthy and costly diagnostic odyssey [94]. Even when covered by insurance, this can delay genetic diagnosis by years.

### Whole-Genome Testing

Although many studies examine the utility and cost-saving potential of WES and WGS in pediatric patients [23,95,96,97], particularly those with neurological and psychiatric disorders, primary literature on the use of WES and WGS in adults is limited [19,95,98,99,100]. The diagnostic utility of WES has already shifted the paradigm of genetic testing in pediatric patient populations, with WES used as a first-tier genetic test in children with certain conditions, rather than a test of last resort [24,25,101,102]. WGS (approaching 100% of the genome) is expected to usurp WES (1% of the genome) as the clinical diagnostic methodology of choice for pediatric patients in the near future [103,104,105,106]. Whether the results of pediatric studies would translate to adult patient populations is unknown. Most undiagnosed adults with complex neurological disorders present with symptoms beginning in adulthood, pointing to different genetic mechanisms and diagnoses. WGS enables the analysis of all known disease-causing genes, as well as the detection of variants missed by WES, such as intergenic and intronic variants, balanced chromosomal rearrangements, structural variants, copy number variants, and repeat expansions, the latter particularly relevant for adult-onset neurogenetic disorders, such as ALS, FXTAS, SCAs, and muscular dystrophies [107]. WGS could potentially shorten the diagnostic odyssey and save significant testing costs for many patients by eliminating the need for multiple, separate, and successive genetic tests with a large cumulative cost; an illustrative example is shown in Figure 1. Additional information on clinical utility can also be reported from WGS data, including medically actionable secondary findings [91], pharmacogenomic variants [108], and carrier status for recessive or X-linked disorders. WGS could potentially offer these opportunities without additional test costs, ultimately also saving healthcare costs. In a study of 382 patients with undiagnosed diseases, 98 (25.7%) obtained a genetic diagnosis through WES or WGS, including 17 through WGS after non-diagnostic WES [109].

## 5. Barriers to Neurogenetic Testing

Unfortunately, payors have lagged behind in recognizing the importance and impact of genetic diagnosis, not only for the patient, but also for their family members [6], and coverage is variable by condition and insurer [110]. Genetic testing remains unaffordable for most patients when it is not covered, with cost concerns being a strong determinant of willingness to undergo testing [111,112,113,114]. In the United States, insurance payor policies on reimbursement and coverage of WES and WGS cite evidence for medical necessity and coverage of WES/WGS in primarily pediatric indications, and lack the same for adult indications, leading to the exclusion of coverage for most adult patients. In the absence of reimbursement or coverage for testing, patient willingness to pay depends on factors beyond clinical utility including personal utility [115]. Personal utility crosses the social, affective, cognitive, and behavioral domains, which are further delineated into fifteen elements that encompass outcomes such as mental preparation, value of information, and change in social support [116,117]. The National Institutes of Health supports developing and applying patient-reported outcome measures to understand patient perspectives and experiences to delineate the full impact of genomic testing [115,117,118,119,120,121,122].

Patients have historically come to genetics services near the end of a diagnostic odyssey, rather than at the beginning of one [105]. This results in patients who are looking for many kinds of answers: what condition they have (or what it is not), how this affects their current and future family members, and how they may need to adapt their identity or psychological well-being to new information. Important aspects that affect the personal utility of genetic testing include the type of genetic testing result, the specific indication of a patient and the potential differential diagnosis, and ultimately whether a patient has a desire for an answer versus the goal to narrow down the differential or rule out genetic diagnosis. Ultimately, personal utility highlights important facets of the lived experience of genetic conditions that are integral when evaluating healthcare outcomes alongside clinical utility and other measures.

While the discussion about the theoretical superiority of WGS over WES has ramped up over the past several years (e.g., [123]), the translation of the hypothetical advantages of WGS into reality has led to the recognition of some of the limitations of “traditional” short-read sequencing and bioinformatic tools available to interpret genomic data. This has led to the development of superior and complementary technologies and tools to overcome some of the technical barriers to realizing the promise of WGS, including long-read sequencing technology, optical genome mapping, multi-omics approaches, and the development of many new bioinformatic tools to allow for interrogation of CNVs, repeat expansions, and non-coding regions [84,124,125,126,127,128,129,130,131].

Even with the advent of new genetic diagnostic tools, most patients have limited access to neurological care [132,133] and/or medical genetics services [2,134,135], and concentrated expertise in rare diseases limits the scope of expedient and accurate diagnosis of adult patients with neurogenetic diseases. Collaboration between neurologists and geneticists is needed to better incorporate clinical and genetic data.

## 6. Conclusions

There is currently scant data to guide adult patients with suspected neurogenetic disorders and their clinicians in making decisions about which genetic testing strategy would bring optimal clinical value to them. The rapid expansion of known pathogenic variants since the advent of WES and WGS has transformed our understanding of neurogenetic disease and disrupted the existing classification schema of these disorders. Increasingly, deep clinical phenotyping in combination with algorithmic approaches of standard neurogenetic testing is not sufficient to achieve accurate genetic diagnoses. Despite the growing impact of genetic testing on clinical practice, most physicians are uncomfortable with the use of medical genetics and genomics, limiting their ability to counsel patients [136,137,138,139,140]. Ongoing collaboration between neurology experts in rare diseases and geneticists is needed to define future guidelines for the diagnosis of rare neurogenetic diseases. Future studies are needed to determine whether a traditional, sequential testing strategy or costlier upfront, but more efficient WES or WGS yields the greatest diagnostic, clinical, and personal utility for adults with complex neurological phenotypes. Studies of this nature in adult populations are urgently needed to enable payor policy changes that will ensure equitable, evidence-based access to the latest genomic technologies for adult patients, as has been widely implemented for pediatric populations.

## Figures and Tables

**Figure 1 biology-12-01459-f001:**
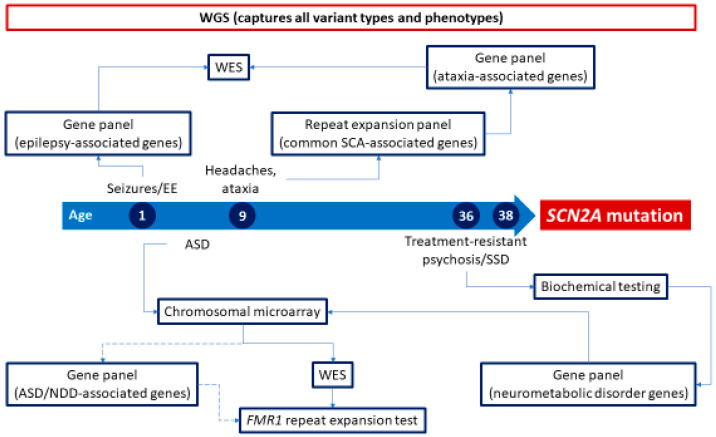
Example of the current state of genetic testing for each phenotype in an adult with complex neurological phenotypes (case based on [12]). The typical diagnostic odyssey entails multiple genetic tests before a neurogenetic disorder can be identified and molecularly confirmed. Access to whole-exome sequencing can often be restricted by insurance/payor coverage policies, limiting testing to gene panels (dotted line) instead of or prior to WES, as illustrated for ASD. Priority should always be to identify treatable genetic disorders, as illustrated by pursuing biochemical and gene panel testing for neurometabolic disorders before other testing for treatment-resistant SSD. Note that WES, in addition to covering only coding regions in the genome, is unable to capture variant types such as repeat expansions and large copy number variants, whereas WGS is able to capture all variant types and be analyzed for all phenotypes. WGS could present a “one-stop shop” test for adult neurogenetic disorders as cost and feasibility improve. Abbreviations: ASD, autism spectrum disorder; EE, epileptic encephalopathy; NDD, neurodevelopmental disorder; SCA, spinocerebellar ataxia; SSD, schizophrenia spectrum disorder; WES, whole-exome sequencing; WGS, whole-genome sequencing.

**Table 1 biology-12-01459-t001:** Recommended first-line genetic testing by neurological symptom.

Clinical Presentation	First-Line Genetic Testing Strategy	Distinctive Clinical Features	Other Considerations
Parkinsonism	Gene panel	Early-onset < 40 years old	
Dystonia	Gene panel	Early age of onset, combined dystonia phenotype	
Chorea	Repeat Expansion Testing Panel +/− Sequencing Panel		Rule out Huntington disease and mimics
Ataxia	Repeat Expansion Testing Panel +/− Sequencing Panel		Repeat expansion should be prioritized if pure SCA (most commonly SCA3, FXTAS); if complex, consider gene panel first
Spastic Paraparesis	Gene panel		
Epilepsy	Gene panel	Seizure onset in infancy, comorbid intellectual disability or developmental delay, pharmacoresistent epilepsy	
Neuropathy	Gene panel	Distal motor and/or sensory neuropathy	Consider sequence evaluation of *PMP22* upstream regulatory elements
CANVAS	Single gene testing	Persistent cough	Novel repeat motifs identified, limited availability of testing
Muscle disorder	Gene panel		If findings strongly suggest a specific muscular dystrophy (e.g., DM, FSHD, OPMD) consider repeat expansion testing
Motor Neuron Disease	Repeat Expansion Testing Panel +/− Sequencing Panel	Association with FTD	Repeat expansion should be prioritized, as most common diagnoses are ALS, SBMA, Friedreich ataxia
Dementia	Gene panel	Early-onset < 65 years old	If FTD, then prioritize repeat expansion (*C9orf72*)
Leukodystrophy	Gene panel		
Cerebral ischemia	Gene panel	Recurrent stroke, early-onset < 50 years old, lack of typical vascular risk factors, symmetric imaging findings	
Episodic Neurological Syndrome	Gene panel		

Abbreviations: ALS, amyotrophic lateral sclerosis; CANVAS, cerebellar ataxia with neuropathy and vestibular areflexia syndrome; DM, myotonic dystrophy; FTD, frontotemporal dementia; FSHD, facioscapulohumeral muscular dystrophy; FXTAS, fragile X-associated tremor/ataxia syndrome; OPMD, oculopharyngeal muscular dystrophy; SBMA, spinal bulbar muscular atrophy; SCA, spinocerebellar ataxia.

## Data Availability

Data sharing is not applicable.

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
