# Peer review of "The Diagnostic Landscape of Adult Neurogenetic Disorders"

_biology, 2023, doi:10.3390/biology12121459_

Round 1

Reviewer 1 Report

Comments and Suggestions for Authors

This review offers a comprehensive overview of the adult onset neurogenetic diseases, pointing out to the factors which up to now caused  long lasting, time and cost consuming , and often unsuccessful search of a definite diagnosis, and how the progresses in the molecular genetic field by NGS techniques  are contributing to facilitate and improve molecular screening, also underlying the limitations that NGS techniques still have in  terms of accessibility and costs to patients .

Below you will find my comments and suggestions to improve the manuscript. Please also check accurately the cited references throughout the manuscript

Line 56. C9ORF72 should be added to the list of HD phenocopies

Line 108: Add a brief sentence which explains that TRP and WES  cannot detect such kind of  rearrangements,  so single gene testing is required

 Line 111 A sentence containing update regarding CANVAS (RFC1-related ataxia  and Fgf14 /SCA27B should be added, as these  genetic variant of ataxias appear to be relatively frequent among degenerative adult ataxias .

Line 118 These inheritance are both very rare so they are not expected to impact  on the overall frequency of HSPS

122. I would suggest to change  This sentence in : This classification is certainly useful , but cannot be assumed as an exclusive  criterion to guide molecular screening of HSPS, as the same mutation….

124 put  a dot after HSPS and change  the following with.. Moreover, WES and WGS diagnostic approaches have revealed a genetic overlap between HSP and other neurogenetic conditions , more commonly inherited ataxias, neuropathies, motor neuron diseases, found to be associated  to genes originally described in HSP patients. In this case, both site and type of the mutation might have distinct pathogenic effects on the gene product, which might explain underlying differences in their clinical presentation.  

130 after the word epilepsy change in  multigene panel tests The majority of patients still have a mutation in one the four most common genes (PMP22 duplication-CMT1A, MPZ-CMT1B, GJB1-CMTX1, and MFN2-CMT2A).

159: “SCAs with neuropathy” are actually already  included in “inherited ataxias with neuropathy”, please remove it

162 . I would suggest to change in ; also phenotypic … … make

172. Remove this sentence, it has been reported in the previous paragraph

179: Add a brief sentence regarding classification of CMT and related numbering  

208: At the beginning of the paragraph , please add Regarding muscle diseases …. to introduce this subgroup of NM disorders

215 Add this sentence please Both disorders are inherited as an autosomal dominant trait ,  and show a wide phenotypic variability, mostly related to the meiotic and mitotic instability in tissues which characterizes related CTG and CCTG  pathological expansions

 217 add , milder allelic variants of the most severe Duchenne muscular dystrophy

218: add the second locus and gene causing  FSDH2 please

Line 223 The list of genetic myopathies with adult onset should also report a sentence reminding also genes related to dystroglycanopathies,  metabolic myopathies , and congenital myopathies

Line 240 Please add a brief sentence reporting the relative percentage of early onset vs late onset AD, indicating that Early onset represent 5-10% dei casi and that to this group belong AD associated with monogenic causes

lIne 258. After imaging. Please add “Most leukodystrophies have the onset of symptoms during childhood or teenage; however, rarer late-onset presentations of such forms may occur ,  whereas, on the other hand, other forms of inherited leukodystrophies are typically characterized by an adult onset “.

Line 272 Add adult-onset  before leukoencephalopathies

Line 301 Add to the list  remethylation disorders ( such as severe MTHFR)

Line 302 I would remove Pompe which is typically a severe infantile muscle glycogenosis, rather citing in its place other lysosomal diseases such HEX deficiency, mannosidosis 

Line 314 Please include in this list also MERRF, NARP , KSS

Line 323, continue the sentence with “whose presence and predominance  in the phenotype also depends on the underlying mutation”

Line 336 Add muscle channellopathies  such as periodic paralysis to this chapter

Line 500 This paper in my opinion also emphasizes the importance of clinicians experts in rare diseases in order to address correct disease, as 10 pts received diagnosis only based on clinical re-evaluation. Alliance between clinicians and genetists is relevant, at least until now .  

IN the conclusions I would suggest to add 1)  a brief sentence commenting on the challenge represented by this review , needing to summarize an issue that , although  limited in terms of number of patients, is still very hard in terms of diagnosis

2) A comment on the fact that, especially in Countries  where insurance  coverages  limit the access to more expensive genetic testing , while waiting for progresses from  AI and WGS technologies to lower  individual costs, an evaluation of patients by tertiary neurological centers with the related expertise still make the difference in terms of diagnostic yields and will also allow to address  research studies useful to define future guidelines for  diagnosis of rare neurogenetic disorders 

Author Response

We appreciate the opportunity to improve our manuscript for publication in Biology. We are grateful to Reviewer 1's thorough review and thoughtful comments. As you can see below, we have incorporated most of the suggestions as recommended. As a result, we feel this revised manuscript is significantly improved, providing a better overview of Neurogenetic disorders for the general Biology audience. Please see specific comments addressed in bold. 

Reviewer 1 

This review offers a comprehensive overview of the adult onset neurogenetic diseases, pointing out to the factors which up to now caused long lasting, time and cost consuming , and often unsuccessful search of a definite diagnosis, and how the progresses in the molecular genetic field by NGS techniques are contributing to facilitate and improve molecular screening, also underlying the limitations that NGS techniques still have in terms of accessibility and costs to patients. 

Below you will find my comments and suggestions to improve the manuscript. Please also check accurately the cited references throughout the manuscript 

Line 56. C9ORF72 should be added to the list of HD phenocopies 

Added. 

Line 108: Add a brief sentence which explains that TRP and WES cannot detect such kind of rearrangements, so single gene testing is required 

This is addressed in the “Current state and challenges of neurogenetic testing” section. 

Line 111 A sentence containing update regarding CANVAS (RFC1-related ataxia and Fgf14 /SCA27B should be added, as these genetic variant of ataxias appear to be relatively frequent among degenerative adult ataxias. 

Added. 

Line 118 These inheritance are both very rare so they are not expected to impact on the overall frequency of HSPS 

Revised to note they are rare. 

  1. I would suggest to change This sentence in : This classification is certainly useful , but cannot be assumed as an exclusive criterion to guide molecular screening of HSPS, as the same mutation….

Revised to note phenotypic pleiotropy of mutations.  

124 put a dot after HSPS and change the following with..Moreover, WES and WGS diagnostic approaches have revealed a genetic overlap between HSP and other neurogenetic conditions ,more commonly inherited ataxias, neuropathies, motor neuron diseases, found to be associated to genes originally described in HSP patients. In this case, both site and type of the mutation might have distinct pathogenic effects on the gene product, which might explain underlying differences in their clinical presentation. 

Added. 

130 after the word epilepsy change in multigene panel tests The majority of patients still have a mutation in one the four most common genes (PMP22 duplication-CMT1A, MPZ-CMT1B, GJB1-CMTX1, and MFN2-CMT2A). 

This comment is unclear to us.  However, it is noted in the CMT section that 90% of CMT is due to mutations in these 4 genes. 

159: “SCAs with neuropathy” are actually already included in “inherited ataxias with neuropathy”, please remove it 

Removed 

162 . I would suggest to change in ; also phenotypic … … make 

This comment is unclear to us. 

  1. Remove this sentence, it has been reported in the previous paragraph

This comment is unclear to us. 

179: Add a brief sentence regarding classification of CMT and related numbering 

Added. 

208: At the beginning of the paragraph , please add Regarding muscle diseases …. to introduce this subgroup of NM disorders 

Added. 

215 Add this sentence please Both disorders are inherited as an autosomal dominant trait , and show a wide phenotypic variability, mostly related to the meiotic and mitotic instability in tissues which characterizes related CTG and CCTG pathological expansions 

Added. 

217 add , milder allelic variants of the most severe Duchenne muscular dystrophy 

Added. 

218: add the second locus and gene causing FSDH2 please 

Added. 

Line 223 The list of genetic myopathies with adult onset should also report a sentence reminding also genes related to dystroglycanopathies, metabolic myopathies , and congenital myopathies 

Dystroglycanopathies are included in limb-girdle muscular dystrophies, which are already mentioned.  Metabolic myopathies are described in the “Neurometabolic diseases” section.  Since congenital myopathies are not adult onset, they are outside the scope of this discussion. 

Line 240 Please add a brief sentence reporting the relative percentage of early onset vs late onset AD, indicating that Early onset represent 5-10% dei casi and that to this group belong AD associated with monogenic causes 

Added. 

lIne 258. After imaging. Please add “Most leukodystrophies have the onset of symptoms during childhood or teenage; however, rarer late-onset presentations of such forms may occur , whereas, on the other hand, other forms of inherited leukodystrophies are typically characterized by an adult onset “. 

Added. 

Line 272 Add adult-onset before leukoencephalopathies 

Added. 

Line 301 Add to the list remethylation disorders (such as severe MTHFR) 

We have added in remethylation disorders and used cobalamin C deficiency as an example. 

Line 302 I would remove Pompe which is typically a severe infantile muscle glycogenosis, rather citing in its place other lysosomal diseases such HEX deficiency, mannosidosis 

While we agree that the infantile form of Pompe disease may present within the first few months of life, the late-onset form of Pompe disease can manifest as late as the 7th decade of life (https://pubmed.ncbi.nlm.nih.gov/21631931/). 

Line 314 Please include in this list also MERRF, NARP , KSS 

Added. 

Line 323, continue the sentence with “whose presence and predominance in the phenotype also depends on the underlying mutation” 

Added. 

Line 336 Add muscle channelopathies such as periodic paralysisto this chapter 

Added. 

Line 500 This paper in my opinion also emphasizes the importance of clinicians experts in rare diseases in order to address correct disease, as 10 pts received diagnosis only based on clinical re-evaluation. Alliance between clinicians and genetists is relevant, at least until now . 

Agree.  We emphasize this in the Simple Summary. 

IN the conclusions I would suggest to add 1) a brief sentence commenting on the challenge represented by this review , needing to summarize an issue that , although limited in terms of number of patients, is still very hard in terms of diagnosis 

Added. 

2) A comment on the fact that, especially in Countries where insurance coverages limit the access to more expensive genetic testing , while waiting for progresses from AI and WGS technologies to lower individual costs, an evaluation of patients by tertiary neurological centers with the related expertise still make the difference in terms of diagnostic yields and will also allow to address research studies useful to define future guidelines for diagnosis of rare neurogenetic disorders 

Included in above and revised Conclusions. 

Reviewer 2 Report

Comments and Suggestions for Authors

I agree with the authors that genetic testing in neurology is an area of rapid growth and a review could be helpful to address physician reservations and knowledge gaps in this area. In its current form, the manuscript reads like an incomplete listing of genetic causes of neurological diseases followed by descriptions of available genetic tests. This manuscript would be improved by focusing the scope of the paper to provide a novel insight. I would additionally point out that contrary to the assertion made in their manuscript, genetic forms of neurodegenerative disease for example are often NOT conserved within families (eg. VCP or C9orf72 etc) and "subacute or chronic progressive unrelenting courses" are a feature of many different neurodegenerative diseases regardless of genetic etiology. 

Author Response

We respectfully disagree with Reviewer's 2 assessment, but appreciate their appraisal of our manuscript. Please see our comments in bold.

Reviewer 2

I agree with the authors that genetic testing in neurology is an area of rapid growth and a review could be helpful to address physician reservations and knowledge gaps in this area. In its current form, the manuscript reads like an incomplete listing of genetic causes of neurological diseases followed by descriptions of available genetic tests. This manuscript would be improved by focusing the scope of the paper to provide a novel insight.  

Thank you.  Since this is a review paper on the current landscape of neurogenetic disorders, the focus is to review broad categories of neurogenetic disorders and testing.  We also emphasize the need to generate novel data to support optimized testing strategies in the future.   

I would additionally point out that contrary to the assertion made in their manuscript, genetic forms of neurodegenerative disease for example are often NOT conserved within families (eg. VCP or C9orf72 etc) and "subacute or chronic progressive unrelenting courses" are a feature of many different neurodegenerative diseases regardless of genetic etiology. 

We have added an additional sentence to highlight the phenotypic pleiotropy that can be associated with certain genes.  We agree that subacute/chronic progressive unrelenting courses are a feature of many different neurodegenerative diseases.  We do not assert that any individual “red flag” is indicative of an underlying neurogenetic disorder.  However, the presence of this finding may, along with other red flags, support the level of suspicion for a neurogenetic disorder and prompt/justify the need for genetic testing. 

Reviewer 3 Report

Comments and Suggestions for Authors

In the present manuscript, the authors have discussed genetic factors behind various neurological disorders, currently available diagnostic methodologies being used, and future developments for more accurate diagnosis with special emphasis on the adult population. The manuscript is comprehensive and up to date. I recommend for the acceptance by the journal with the following minor improvements-

1- Please include high-quality picture.

2- Minor English check required.  

Comments on the Quality of English Language

English quality is good. Only a minor check is required.

Author Response

We appreciate Reviewer's 3 comments. Please see our responses in bold.

Reviewer 3 

In the present manuscript, the authors have discussed genetic factors behind various neurological disorders, currently available diagnostic methodologies being used, and future developments for more accurate diagnosis with special emphasis on the adult population. The manuscript is comprehensive and up to date. I recommend for the acceptance by the journal with the following minor improvements- 

  1. Please include high-quality picture. 

Please advise if this higher-quality picture is needed, as this figure was also uploaded separately.  

  1. Minor English check required. 

We have reviewed the article carefully for English errors, and have corrected mistakes. We appreciate any additional copy editing advice that may have missed our review.